# *Phytobacter diazotrophicus* from Intestine of *Caenorhabditis elegans* Confers Colonization-Resistance against *Bacillus nematocida* Using Flagellin (FliC) as an Inhibition Factor

**DOI:** 10.3390/pathogens11010082

**Published:** 2022-01-10

**Authors:** Qiuhong Niu, Suyao Liu, Mingshen Yin, Shengwei Lei, Fabio Rezzonico, Lin Zhang

**Affiliations:** 1College of Life Science and Agricultural Engineering, Nanyang Normal University, 1638 Wolong Road, Nanyang 473061, China; qhniu@nynu.edu.cn (Q.N.); suyaoliu062@163.com (S.L.); yms13944991620@163.com (M.Y.); shengweilei12@163.com (S.L.); 2Environmental Genomics and Systems Biology Research Group, Institute of Natural Resource Sciences, Zurich University of Applied Sciences (ZHAW), 8820 Wädenswil, Switzerland

**Keywords:** *Phytobacter diazotrophicus*, colonization-resistance, flagellin FliC, *C. elegans*, *Bacillus nematocida* B16

## Abstract

Symbiotic microorganisms in the intestinal tract can influence the general fitness of their hosts and contribute to protecting them against invading pathogens. In this study, we obtained isolate *Phytobacter diazotrophicus* SCO41 from the gut of free-living nematode *Caenorhabditis elegans* that displayed strong colonization-resistance against invading biocontrol bacterium *Bacillus nematocida* B16. The colonization-resistance phenotype was found to be mediated by a 37-kDa extracellular protein that was identified as flagellin (FliC). With the help of genome information, the *fliC* gene was cloned and heterologously expressed in *E. coli*. It could be shown that the *B. nematocida* B16 grows in chains rather than in planktonic form in the presence of FliC. Scanning Electronic Microscopy results showed that protein FliC-treated B16 bacterial cells are thinner and longer than normal cells. Localization experiments confirmed that the protein FliC is localized in both the cytoplasm and the cell membrane of B16 strain, in the latter especially at the position of cell division. ZDOCK analysis showed that FliC could bind with serine/threonine protein kinase, membrane protein insertase YidC and redox membrane protein CydB. It was inferred that FliC interferes with cell division of *B. nematocidal* B16, therefore inhibiting its colonization of *C. elegans* intestines in vivo. The isolation of *P. diazotrophicus* as part of the gut microbiome of *C. elegans* not only provides interesting insights about the lifestyle of this nitrogen-fixing bacterium, but also reveals how the composition of the natural gut microbiota of nematodes can affect biological control efforts by protecting the host from its natural enemies.

## 1. Introduction

The gut microbiota play important roles in the host fitness by influencing metabolism, development, immunity, and lifespan [1]. The intestine of free-living soil nematode *Caenorhabditis elegans*, which is formed by 20 epithelial cells, is typically full of microbes [2]. *C. elegans* is considered a ‘microbivore’ due to its feeding behavior that consists of eating living bacteria. Soil bacteria can be digested providing nutrients to the nematodes or can survive in their intestinal tract, establishing a complex relationship that can range from symbiotic to pathogenic [3,4,5,6]. Due to its easy cultivation, simple structure and genetically tractable characteristics, *C. elegans* has become a preferred subject for studying the interactions between host and bacteria [7]. Previous studies have identified the normal intestinal flora within worms that maintain these stable symbiotic relationships [2,8]. Félix and Duveau reported the presence of bacteria in wild *C. elegans* but did not identify the species present [2], while Montalvo-Katz and coworkers identified some of the species present in the intestine of *C. elegans* and examined whether these can confer resistance to nematode pathogens *Pseudomonas aeruginosa* and *Enterococcus faecalis* [8]. Accumulating evidence indicates that some gut symbiotic bacteria antagonize pathogenic bacteria [9]. Life-history traits, such as development, reproduction, and lifespan are altered in *C. elegans* when fed with different bacterial species [10]. Further, symbiotic bacteria influence the immune system in *C. elegans* and contribute to resistance against invading bacteria. For instance, worms fed with lactic-acid-producing bacteria are more resistant to *Salmonella enterica* than those fed with the *E. coli* strain OP50 [9]. *Pseudomonas mendocina* enhances innate immune responses to Gram-negative pathogens in *C. elegans* [8], while *Bacillus subtilis* GS67 confers resistance to the Gram-positive pathogens in *C. elegans* [11]. In soil, pathogenic bacteria are natural enemies of nematodes, and were thus proposed as biological control agents [12]. According to their modes of action against nematodes, nematophagous bacteria are classified into the following groups: obligate parasitic bacteria, opportunistic parasitic bacteria, rhizobacteria, parasporal Cry protein-forming bacteria, endophytic bacteria, and symbiotic bacteria [13]. Organisms belonging to the genera *Bacillus*, *Pseudomonas*, and *Pasteuria* represent the dominant populations of nematophagous bacteria in soil [14]. For the successful control of nematodes, a sufficient population density of invading bacteria is needed. However, symbiotic bacteria can strongly repress the pathogenic bacteria used as biocontrol agents, making them inefficient [15]. The phenomenon of intestinal microbiota effectively inhibiting colonization and overgrowth by invading microbes is called “colonization-resistance” [16].

Flagella, which consists of multimers of flagellin, are the organelles responsible for mobility in many bacteria and are important factors influencing their lifestyle in different environments [17]. As the primary component of bacterial flagella, flagellin (FliC) has been reported to act as immunogenicity factor in different studies [18,19,20,21]. FliC has also been reported as a virulence factor that contributes to the adhesion and invasion of host cells [22]. FliC from opportunistic marine pathogen *Vibrio splendidus* AJ01 contributes to bacterial adhesion and colonization in intestines of their animal hosts [23]. Suriyanarayanan et al. found that FliC phosphorylation had a role in ecological adaptation of *Pseudomonas aeruginosa* PAO1, which affects key surface-related processes such as proteases secretion by the type 2 secretion system, biofilm formation and dispersal [17]. On the other hand, plants have evolved receptors to sense the FliC protein as signatures of bacterial infection. Bao et al. found that *Pseudomonas syringae* protein AlgU downregulates flagellin gene expression, helping the bacteria to evade plant defenses by reducing the elicitation of the plant immune system [24]. These results suggest that FliC may function as a key control point to regulate bacterial virulence functions during interaction with the host.

In our previous study we identified the “normal” intestinal microflora of adult *C. elegans* and found that regardless of the worm’s origin (i.e., from soil or rotten fruits) the diversity of the intestinal microbiome decreased after infection by biocontrol bacterium *Bacillus nematocida* B16 [25]. Here, we purified the putative inhibition factor of *Phytobacter diazotrophicus* SCO41 and identified it as FliC, one of the protein subunits that form the bacterial flagella. The role of FliC in against *B. nematocida* B16 was verified both by using the isolated protein or by cloning and heterologous expression in *E. coli*.

## 2. Results

### 2.1. Identification Analysis

Of the 52 bacterial isolates obtained from the intestines of *C. elegans* twelve of them had the ability to antagonize the colonization by the invading bacterium *B. nematocida* B16 [26]. Multilocus sequence analysis based on *gyrB*, *rpoB*, *atpD* and *infB* genes of the three *Phytobacter* isolates resulted in undistinguishable sequences and allowed refining their identification to *P. diazotrophicus* (Appendix A). The other two taxa identified were *Pediococcus* sp. (4 isolates) and *Escherichia coli* (5 isolates) [26]. The strain *P. diazotrophicus* SCO41 with the strongest inhibition activities against B16 both in vivo and in vitro was selected as the target bacteria in following research.

### 2.2. Purification of the Inhibition Factor and Identification

The biological activity was determined by plate inhibition assay. That is, the filter paper was soaked with untreated, boiled, and 0.1-M proteinase K treated *P. diazotrophicus* SCO41 culture supernatant, and the LB liquid was used as a blank control. After being drained, the filter paper was placed on a plate coated with *B. nematocida* B16 and incubated in a constant temperature incubator at 37 °C for 24 h to observe the inhibitory effect. Through the plate inhibition assay, it was found that the culture supernatant of the *P. diazotrophicus* SCO41 presented inhibition capabilities against *B. nematocida* B16 while boiled and proteinase K treated culture supernatant showed no activity, thus suggesting the involvement of proteins (Figure 1A). Purification of proteins with colonization-resistance activity from culture supernatant of *P. diazotrophicus* SCO41 was performed by gradient ammonium sulfate precipitation. The bioactivity was assayed in each fraction by the plate inhibition assay. The component under 30–40% gradient was detected as containing a substance inhibiting *B. nematocida* B16 in vitro (Figure 1B). The purified protein also showed obvious inhibition against B16 growth and the inhibition activity was positively correlated with the amount of protein (Figure 1C). Moreover, SDS–PAGE of the corresponding fraction showed a single protein band with a molecular mass about 37 kDa (Figure 1D). To identify the protein, purified band from *P. diazotrophicus* SCO41 were excised from the SDS-PAGE gel and subjected to in-gel trypsin digestion and subsequent MALDI-TOF/TOF identification. According to the N-terminal sequence alignment analysis and annotations from protein library including UniProt knowledgebase (Swiss-Prot/TrEMBL) and Gene Ontology Database, the protein may be identified to be FliC (flagellin), the subunit protein that polymerizes to form the filaments of bacterial flagella.

### 2.3. Cloning of the Gene and Sequence Analysis

Based on its measured molecular mass, the FliC protein was identified as the possible inhibition factor produced by *P. diazotrophicus* SCO41. The gene encoding FliC in its genome was cloned and sequenced. The deduced protein sequence is composed of 288 amino acids and was submitted to GenBank (WP_108701038). The phylogenetic tree based on amino acid sequence was constructed to verify the evolutionary relationship of FliC to other known flagellins, and 25 closely related FliC protein sequences were selected for distance analysis (Figure 2). The amino acids residues showed the highest identity (97.59%) with flagellin FliC from ‘*Metakosakonia* sp.’ MRY16-398 and cluster with a group of strains that, in spite of the names under which the sequences were deposited, also belong to *P. diazotrophicus*. As a matter of fact, based on genomic comparisons, a recommendation to unite *Metakosakonia* and *Phytobacter* under the latter genus name was recently proposed by Ma et al. [27] and misidentification of *Phytobacter* isolates as *Klyuvera, Citrobacter* or ‘*Grimontella*’ has been reported before [28]. It is interesting to note that the genus *Phytobacter* does not appear to be monophyletic with respect to FliC, as other clusters contain a mix of several *Phytobacter* species and are interspersed among sequences belonging to other genera (Figure 2).

### 2.4. Heterologous Expression and Purification of rm-FliC

The complete coding region of the deduced *fliC* gene was inserted into the heterologous expression plasmid pET32a and transfected into competent cells of *E. coli* BL21. After optimization, the best induction conditions are with 0.6 mM IPTG at 25 °C for 12 h (Figure 3A). The recombinant protein, termed rm-FliC, was purified using a Ni-NTA column (Qiagen company). SDS-PAGE analysis showed an approximate 53 KDa protein band as soluble form (Figure 3B). This molecular weight is consistent with mature rm-FliC fused to the His-tag coupled with thioredoxin.

### 2.5. Bioassay and Localization Results Using rm-FliC Protein

After the proteins FliC or rm-FliC were added into LB medium containing *B. nematocida* B16 for overnight culture, it was found that although the A_660_ of *B. nematocida* B16 cultures was not different with that of control, the cell morphology was noticeably altered. The *B. nematocida* B16 cells treated with purified protein FliC and rm-FliC were both arranged in long chains (Figure 4B,C), while the normal *B. nematocida* B16 as well as the control samples treated with PBS group (Figure 4A), were presenting themselves in a normal dispersed form. Additionally, the results of the SEM observations showed that compared with normal cells (Figure 5A), the protein-treated cells were thinner and longer, and some of them were bent (Figure 5B,C). The results indicated that the protein FliC in *P. diazotrophicus* SCO41 may affect the normal cellular division of *B. nematocida* B16.

The results of automated quantitative fluorescence analysis (Figure 6D), which combined FITC-antibody fluorescence intensity to locate the site of action of the rm-FliC protein in *B. nematocida* B16 (Figure 6B) and DAPI staining to confirm the position of the cells instead of other impurities (Figure 6C), demonstrated that, while the flagellar proteins were generally distributed throughout the cell, some of the latter showed significant accumulation at the points of cell division, as indicated by the arrows in Figure 6B. These data suggest that, while the rm-FliC protein possibly acts both on the cell membrane and cytoplasm, it plays a major role in interfering with the normal cell division process of *B. nematocida* B16, thus reducing its colonization ability and fitness.

The results of the inhibition colonization assay in vivo showed that *B. nematocida* B16 amount and fluorescence intensity of the reporter strain *B. nematocida* B16g in worm intestines were decreased when the worms were exposed to proteins FliC and rm-FliC before infection (Figure 7A,B). Moreover, a significant decrease in worm mortalities was observed throughout the assay compared with those PBS buffer as the control group (Figure 7C). However, protein FliC alone has slightly lower colonization inhibition activity compared to that obtained with the *P. diazotrophicus* SCO41. This suggests that the resistance colonization conferred by *P. diazotrophicus* SCO41 is mediated by flagellin FliC, but that the latter is probably not the only inhibition factor.

### 2.6. Identification and Quantitation of Differentially Expressed Proteins after rm-FliC Treatment In Vitro

The iTRAQ analysis of differential protein expressions in the control and the experimental group of *B. nematocida* B16 treated with rm-FliC was performed. To estimate the false positive rate for protein identification, all spectra were searched with all Swiss-Prot sequences reversed. Only proteins that were present at a false discovery rate (FDR) lower than 1% and identified by at least two peptides were considered. A total of 1142 proteins with significantly different protein expression were identified when considering both sampling points, including 585 with significantly up-regulated protein expression and 557 with significantly down-regulated protein expression. There were 203 and 506 up-regulated proteins at 4 h and 24 h, respectively. Among them, 124 proteins were simultaneously upregulated at both time points (Figure 8A). These up-regulated proteins can be divided into five categories: ribosomal proteins (27%), dehydrogenases (12%), transferases (9%), elongation factors (4%) and other proteins (48%) (Figure 8B). The down-regulated proteins were mainly related to biological processes such as metabolism and stress response. There were 16 and 541 down-regulated proteins treated with rm-FliC for 4 h and 24 h, respectively. However, there was no protein that was down-regulated at both time points (Figure 8C). The down-regulated proteins can be divided into six categories: spore-related proteins (23%), transferases (6%), dehydrogenases (6%), peptidases (5%), binding proteins (5%) and other proteins (55%) (Figure 8D).

According to annotations from UniProt knowledgebase (Swiss-Prot/TrEMBL), GO database and PANTHER database, the identified differentially expressed proteins were categorized into cell components including ribosomal and membrane proteins, molecular function proteins related to catalytic and binding activities, and proteins involved in biological process including expression, metabolism and stress response. After *B. nematocida* B16 was treated with rm-FliC for 24 h, the expression of stress proteins including spore germination, sporulation, and oxidative stress-related stress proteins decreased significantly. KEGG pathway analysis revealed that the differential proteins were involved in glycolysis, infection, HIF-1 signaling pathway and two-component system process. The two-component signal transduction system initiates cell responses by receiving external stimuli to regulate the life activities of bacteria, including bacterial growth, division, spore formation, osmotic pressure balance, and so on. With the prolongation of rm-FliC treatment of *B. nematocida* B16, the protein expression related to the two-component system was down-regulated (Figure 9). Stress tolerance assays results demonstrated that the growth rate of the B16 strain in presence of rm-FliC was significantly lower than that of the untreated control strain under heat, drought and hydrogen peroxide pressure (Appendix A), which demonstrated that *B. nematocida* B16 cells with FliC may exhibit lower tolerance to heat, dryness, and H_2_O_2_ than the control cells. The results indicate that FliC may inhibit the activity of *B. nematocida* B16 by reducing its ability to adapt to external stresses, thereby achieving the purpose of inhibiting nematode colonization.

### 2.7. Molecular Docking Analysis

The output results of ZDOCK analysis indicated that the scores of the membrane protein insertase YidC, a serine/threonine protein kinase, and micro-aerobic cytochrome bd terminal oxidase CydB with FliC are 1945.839, 1668.462, and 1994.676, respectively. The predicted binding complexes were shown in Figure 10 via analyzing the hydrophobic, hydrogen bond and electrostatic interaction.

There are many important amino acid residues at the complex model binding interface of three potential receptors. The detail amino acid residue sequences of the interactions between FliC and the three receptors are listed in Appendix A.

The above analysis revealed that FliC could interact with the amino acid residues at the binding interface of the three receptors, respectively, which will lay the foundation for the subsequent study of the molecular mechanism of protein FliC at specific sites in *B. nematocida* B16.

## 3. Discussion

Using polyphasic evidence generated through phenotypic, chemotaxonomic, and genotypic characteristics, we were able to refine the classification of strain SCO41 (formerly *Phytobacter* sp.) and to positively assign it to *P. diazotrophicus*, a nitrogen-fixing enterobacterium that has its natural habitat in soil. The species was first established as an endophyte of wild rice in China [29] but has also been found associated to the roots of other plants such as sugarcane [30], switchgrass and date palm in Colombia [31], the USA and the United Arab Emirates, respectively. However, *P. diazotrophicus* does not live exclusively in plant tissue as it could also be retrieved as a free-living organism in sandy soil [32]. Considering its ecological niche, it is not surprising that *P. diazotrophicus* can interact with other soil organisms such as nematodes. The presence of *P. diazotrophicus* in the intestinal tract of animals is not a novelty, as the bacterium was, in many instances, previously isolated from the human gut or in stool samples [28,33,34], showing its ability to thrive in this environment. The existence and the type of such interaction in nematodes was, however, so far unknown. In this work, we expand the known lifestyle of *P. diazotrophicus* by demonstrating its probiotic role as a part of the gut microbiome of *C. elegans*, showing its capability to antagonize colonization by nematopathogenic bacteria. 

We were further able to show that flagellar filament structural protein FliC is secreted by *P. diazotrophicus* SCO41 and plays a major role in colonization-resistance by preventing the establishment of *B. nematocida* B16 in worms. FliC was previously found to play an important role in the regulation of innate and adaptive immunity of the animal hosts [35,36]. In *Aeromonas caviae*, FliC is involved in pathogenicity by contributing to the biofilm production [37], while in *Methanothermobacter thermautotrophicus* it participates to the mediation of symbiosis via the flagellum [38]. It is remarkable that, under the investigated conditions and contrary to the species description [29], *P. diazotrophicus* SCO41 does not exhibit any evident flagellar structure according to morphological observations. This may be an indication of a defective assembly mechanism that results in the release of FliC in the extracellular medium where it acts as colonization-resistance factor. In this perspective, further studies would be required to investigate, whether other *P. diazotrophicus* isolates that display normal amphitrichous flagella can provide comparable levels of protection against *B. nematocida* B16.

According to the output of ZDOCK, it was found that the ZDOCK scores of the docking of FliC with serine/threonine protein kinase, membrane protein insertase YidC, and redox membrane protein CydB were higher than other tested receptor proteins. Studies have shown that serine/threonine protein kinase plays an important role in growth, DNA replication, sporulation and germination, stress response and adaptive response [39]. It can also participate in cell shape and cell division control [40]. Membrane proteins are very important to cell functions, such as changes in the external environment, transportation of nutrients, redox balance, cell defense, etc. [41]. YidC is involved in the biogenesis of membrane proteins [42]. In bacteria, YidC mediates the insertion and assembly of membrane proteins related to Sec transporters (translocases) and can also act as an independent factor during membrane insertion of protein [8]. FliC may combine with kinases and membrane proteins in *B. nematocida* B16 and affect bacterial growth, division and cell function, thereby inhibiting the colonization of *B. nematocida* B16 nematodes in the intestine. Redox membrane protein CydB is a cytochrome bd terminal oxidase often present in organisms grown with a limited oxygen supply [43], a condition that is common to gut bacteria. Interference in the electron transport chain across the membrane impacts bacterial respiration under micro-aerobic conditions, thus possibly leading to an important loss of fitness and pathogenicity in *B. nematocida* B16, a mechanism that was documented before in *Brucella abortus* [44].

FliC is, however, not the only factor involved in this process as demonstrated by the fact that the purified protein alone has lower colonization-resistance activity compared with that obtained with the living *P. diazotrophicus* SCO41. In our previous study, we found that enterobactin plays a key role in *P. diazotrophicus* SCO41 ability to oppose the colonization of nematodes by *B. nematocida* B16 via a mechanism involving competing for iron [26]. Additionally, genome analysis showed that the *csgABCEFG* gene cluster is present in *P. diazotrophicus* SCO41 and that the related curli adhesive process may contribute to the colonization of *C. elegans* [26], thus allowing the long-term establishment of *P. diazotrophicus* SCO41 in the gut microbiome of the nematode. Curli and flagella both belong to adhesion factors of bacteria, which mediate key steps in the interaction between bacteria and their hosts.

There have been similar reports on the resident intestinal microbes acting as a barrier against invading microbes. For example, Rangan and Hang reported the biochemical mechanisms by which intestinal bacteria interact with other bacteria and host pathways to restrict pathogen infection [30]. Davoodi and Foley studied on the crosstalk between the enteric pathogen *Vibrio cholerae* and host microbes using *Drosophila melanogaster* as a model. They found that *V. cholerae* could compete with intestinal microbes to improve *Vibrio* fitness [31]. Understanding the inhibition mechanism of intestinal microbiota could provide new opportunities for therapeutic development towards a variety of infectious diseases caused by invading bacteria and also has potential application value in the biological control of pathogenic nematodes.

The enhancement of the nematicidal effect of antagonist bacteria is vital for development of biocontrol agents like *B. nematocida* B16. Successful colonization of the host intestines is a crucial factor to complete infection and attain maximum biocontrol efficiency. Avoiding the protection provided by gut symbiotic bacteria is the key to improve the effects of biocontrol but can only be achieved with an adequate understanding of the underlying molecular mechanisms.

## 4. Methods

### 4.1. Isolation, Screening and Culture Condition

The wild living nematodes were isolated using the Baerman funnel technique [45] and the nematode species was identified by diagnostic PCR using the primer pair nlp30 diagnostic for *C. elegans* [46]. Approximately 50 nematodes with resistance to the infection by *B. nematocida* B16 were washed three times with M9 buffer, and the worms were surface-sterilized by soaking in a mixture of 1% mercuric chloride and 2% antibiotic containing streptomycin sulfate and gentamicin for 1 h, and then cultured on nutrient and oligotrophic agar plates to verify the success of the procedure, which was confirmed if no colony forming units were found. The tested worms were frozen with liquid nitrogen and ground for 10 min with a 1-mL micro-dismembrator (Wheaton), plated on LB agar medium and cultured at 37 °C overnight. The bacteria in the intestine were then identified based on a polyphasic taxonomic method. All of the experiments above were carried out in three replicates and conducted twice. *P. diazotrophicus* SCO41 was one of fifty-two isolates recovered from ten different batches of *C. elegans* samples collected from different places in Nanyang (Henan Province, Central China) [26] and single colonies were purified by the conventional streaking technique on LB agar plates. Purified bacterial isolates were suspended in LB broth supplemented with 10% glycerol and maintained at −80 °C. The bacteria were grown on LB medium for phenotypic, chemical and molecular systematic studies at 37 °C for 2–3 days.

The opportunistic pathogen *B. nematocida* B16 strain was retrieved from China General Microbiological Culture Collection Center (CGMCC 1128) [47]. GFP-expressing *B. nematocida* strain B16g was constructed in our previous study [48].

The growth, synchronization, collection and surface sterilization of the worm *C. elegans* were performed as follow: First, *C. elegans* was grown on solid standard nematode growth medium (NGM) plates at 25 °C and fed *E. coli* OP50 using water-soluble cholesterol. The worms were then separated from the bacteria by sedimentation and sucrose flotation. Eggs were obtained by incubating mixed-stage populations with alkaline hypochlorite. Synchronous cultures were achieved by allowing the purified eggs to hatch overnight in S medium without bacteria. *C. elegans* were grown on standard nematode growth medium plates seeded with the *E. coli* strain 109 g expressing *gfp* as the food source at 25 °C for 24 h. Then, the nematodes synchronized to the L4 stage were selected as following experiments. The different stages of *C. elegans* were verified by visual inspection under a microscope. Adult worms of the same size were harvested and selected for the subsequent experiments. *C. elegans* were washed thoroughly with sterile water before using in the assays [25,48,49].

The assays of colonization-resistance were implemented as follow based on the literature by Aballay et al. with modifications [50]. Approximately 50 surface-sterilized adult hermaphrodite worms were first fed on an isolated gut bacterium lawn for 6 h and then removed from the plates, washed twice in M9 buffer, and then transferred to *B. nematocida* B16g plates for 3 days. To assess whether the isolated bacteria confer colonization-resistance, the fluorescence of the reporter strain *B. nematocida* B16g, the number of colonizing *B. nematocida* B16 bacteria within the *C. elegans* digestive tract and the number of worms killed was separately measured every 12 h using the method described in our previous report [49]. Separate nematode groups not fed on gut bacteria but on *E. coli* OP50 were used as negative controls. The experiments were performed with three parallel replicates and repeated three times.

### 4.2. Purification of the Inhibition Factor and Mass Spectrometry

A 500-mL culture of *P. diazotrophicus* SCO41 was centrifuged at 8000 rpm for 15 min to remove bacteria and obtain supernatant. The supernatant was subjected to gradient ammonium sulfate saturation (0–20%, 20–30%, 30–40%, 40–60%, 60–90%) by slow continuous stirring at 4 °C. After dialyses to desalt, each fraction was tested for inhibiting *B. nematocida* B16 activities. Sodium dodecyl sulfate (SDS)—12% (*w*/*v*) polyacrylamide gel electrophoresis (PAGE) was performed to check the purified protein.

The protein slices were manually excised from the stained gels and then transferred to V-bottom 96-well microplates loaded with 100 μL of 50% Acetonitrile (CAN)/25 mM ammonium bicarbonate solution per well. After destaining for 1 h, gel plugs were dehydrated with 100 μL of 100% ACN for 20 min and then thoroughly dried in a SpeedVac concentrator (Thermo Savant, Waltham, MA, USA) for 30 min. The dried gel particles were rehydrated at 4 °C for 45 min with 2 μL/well trypsin (Promega, Madison, WI, USA) in 25 mM ammonium bicarbonate, and then incubated at 37 °C for 12 h. After trypsin digestion, the peptide mixtures were extracted with 20 μL extraction solution (50% ACN/0.5% TFA) per well at 37 °C for 1 h. Finally, the extracts were dried under the protection of N_2_.

The peptides were eluted with 0.8 μL matrix solution (α-cyano-4-hydroxy-cinnamic acid (CHCA, Sigma, St. Louis, MO, USA) in 0.1% TFA, 50% ACN) before spotted on the target plate. Samples were allowed to air-dry and analyzed by 5800 MALDI-TOF/TOF Proteomics Analyzer (Applied Biosystems, Foster City, CA, USA). MALDI-TOF/TOF MS and MS/MS analysis and database search was completed with the help of GeneCore Company (Shanghai, China).

### 4.3. Cloning of the Gene Encoding the Inhibition Factor

Oligonucleotide primers evtl-f (5′-ATGGCAGTTATCAATACTA-3′); and evtl-r (5′-ACGCAGCAGAGACAGAACA-3′) were designed based on the genome sequence information of *P. diazotrophicus* SCO41 combined with the results of mass spectrometry of the inhibition factor. The PCR product of 867 bp was inserted into pMD18-T vector (Promega) and sequenced for identification. The deduced amino acid sequence was analyzed with BLAST program provided by NCBI. The twenty-five amino acid sequences with the closest homology were selected to construct the phylogenetic tree from Kimura’s two-parameter model [51] using the neighbor-joining method [52].

### 4.4. Heterologous Expression and Purification of FliC in P. diazotrophicus SCO41

The *fliC* gene was amplified by PCR with an EcoRI-linked forward primer Eevtl-f (5′-CGGAATTCATGGCAGTTATCAATACTA-3′) and a HindIII-linked reverse primer Hevtl-r (5′-CCAAGCTTACGCAGCAGAGACAGAACA-3′). Amplified DNA was digested by EcoRI /HindIII, ligated into pET-32a (Promega) and then transformed into *E. coli* BL21 cells after sequence confirmation. Transformed cells were then grown at 37 °C in Luria-Bertani (LB) medium containing kanamycin (50 µg/mL) to a cell destiny of A_660_ = 0.6. Protein expression was induced by 0.6 mM isopropyl-β-D-thiogalactopyranoside (IPTG) (Sigma) at 25 °C for 12 h. The transformant cells were harvested by centrifugation at 12,000× *g* for 5 min, and then the cells were suspended in 5 mL cell lysis buffer and sonicated 50 times at 30 kHz for 5 s in an ice-water bath. The recombinant proteins designated rm-FliC was purified using purification the protocol of Ni-NTA affinity chromatography and tested for bioactivities.

### 4.5. Bioassays Using the FliC Protein

Bioactive assays of the FliC protein include antagonizing growth assay of *B. nematocida* B16 in vitro and inhibiting colonization assay against *B. nematocida* B16 in *C. elegans* intestines. The antagonistic growth assay of *B. nematocida* B16 in vitro: 50 µL 10 µg/mL purified FliC and recombinant FliC (rm-FliC) were separately added into 5 mL LB medium, and an equal volume of PBS buffer added into LB medium was used as negative control group. Then *B. nematocida* B16 was inoculated to the three groups of mixed solution, respectively. After culturing at 37 °C for 12–16 h (overnight) under shaking at 200 rpm, the growth status, cell and colony morphology of *B. nematocida* B16 were observed by optical microscope and scanning electron microscope (SEM) examination (Philips XL30). The inhibition of colonization assay against *B. nematocida* B16 in vivo: 50 µL 10 µg/mL protein FliC were spread onto the NGM plate for culturing worms at room temperature. Then the pathogenic bacteria *B. nematocida* B16g were used to infect nematodes, and the intestinal fluorescence and the number of *B. nematocida* B16 colonization in intestinal tract were measured to evaluate the in-vivo activity of protein FliC. The groups added with the same amount of PBS (pH 7.0, sterile) were used as negative controls in the above experiments. The experiment was repeated three times.

### 4.6. Localization of the FliC Protein in B. nematocida B16

A his-tag antibody was used to specifically label rm-FliC. FITC-Antibody conjugation of rm-FliC was as follows: Firstly, the purified rm-FliC was dialyzed with sodium bicarbonate buffer. The protein concentration was detected by micro-ultraviolet-visible spectrophotometer. Then, the antibody buffer was dissolved with FITC by DMSO and added 1% proportion of FITC. The FITC-conjugated antibody was dialyzed after being added into the glycine. Finally, the FITC-conjugated antibody was detected by micro-ultraviolet-visible spectrophotometer.

*B. nematocida* B16 was cultured in suspension into the logarithmic growth phase. Then, 0.8 mL of the bacterial suspension were transferred in a sterilized centrifuge tube, and 200 μL FITC-P1 were added followed by incubation at 37 °C and 200 rpm for 24 h. The protein buffer was used as the negative control. After washing the bacteria, a final concentration of 0.02% Triton-X100 was used for cell permeability treatment. A 4% paraformaldehyde solution was applied to *B. nematocida* B16 to fix the cells, which were then dyed with DAPI to further demonstrate the B16 cell positions. The stained bacteria were examined on a glass slide under a Nikon 800 microscope equipped for epifluorescence with a mercury lamp. FITC fluorescence was visualized with an excitation filter at 492 nm and a barrier filter at 520 nm. The absorption spectrum of DAPI displayed a strong peak located at 358 nm and a barrier filter of 461 nm.

### 4.7. Proteomic Analysis

Through antagonistic growth assay, it was found that when *B. nematocida* B16 was treated with FliC for 4 h, the cell morphology of *B. nematocida* B16 was undergoing some morphological changes that were most obvious at 24 h. Therefore, 4 h and 24 h were selected for iTRAQ (isobaric tags for relative and absolute quantitation) proteomic analysis.

Trypsin digestion and iTRAQ labeling were performed according to the manufacturer’s protocol (Applied Biosystems, Foster City, CA, USA). Briefly, 100 μg protein of each sample was reduced and alkylated, and then digested overnight at 37 °C with trypsin (mass spectrometry grade; Promega, Madison, WI, USA) and labeled with iTRAQ™ reagents (Applied Biosystems) as follows: control group for 4 h, iTRAQ reagent 117; control group for 24 h, iTRAQ reagent 118; FLIC treatment group for 4 h, iTRAQ reagent 119; and FliC treatment group for 24 h, iTRAQ reagent 121. The sample refers to *B. nematocida* B16 treated with PBS buffer and rm-FliC, respectively. The rm-FliC treatment group refers to *B. nematocida* B16 treated with rm-FliC, and the control group refers to treatment with the same amount of PBS buffer. The labeled samples were combined, desalted using a C18 SPE column (Sep-Pak C18, Waters, Milford, MA, USA) and dried in vacuum.

The peptide mixture was redissolved in buffer A (10 mM ammonium formate in water, pH 10.0, adjusted with ammonium hydroxide), and then fractionated by high pH separation using a Aquity UPLC system (Waters Corporation, Milford, MA, USA) connected to a reverse phase column (BEH C18 column, 2.1 mm × 150 mm, 1.7 μm, 300 Å, Waters Corporation, Milford, MA, USA). High pH separation was performed using a linear gradient. Starting from 0% B to 45% B in 35 min (B: 10 mM ammonium formate in 90% ACN, pH 10.0, adjusted with ammonium hydroxide). The column flow rate was maintained at 250 μL/min and column temperature was maintained at 45 °C. Twelve fractions were collected, each fraction was dried in a vacuum concentrator for the next step.

The peptides were dissolved in 0.1% formic acid (solvent A), directly loaded onto a reversed-phase analytical column (Acclaim PepMap C18, 75 μm × 50 cm). The gradient was comprised of an increase from 2% to 30% solvent B (0.1% formic acid in 98% acetonitrile) over 110 min, 30% to 50% in 5 min and climbing to 80% in 1 min then holding at 80% for the last 4 min, all at a constant flow rate of 200 μL/min on an EASY-nLC 1200 UPLC system. The peptides were subjected to NSI source followed by tandem mass spectrometry (MS/MS) in Orbitrap Exploris 480 MS coupled online to the UPLC. Spray voltage were set to 2.3 kV, funnel RF level at 50, and heated capillary temperature at 320 °C. For DDA experiments full MS resolutions were set to 60,000 at *m*/*z* 200 and full MS AGC target was 300% with an IT of 50 ms. Mass range was set to 450–1800. AGC target value for fragment spectra was set at 200% with a resolution of 30,000 and injection times of 50 ms and Top12. Intensity threshold was kept at 2 × 10^5^. Isolation width was set at 1.6 *m*/*z*. Normalized collision energy was set at 35%.

Tandem mass spectra were extracted by Proteome Discoverer software (Thermo Fisher Scientific, Waltham, MA, USA, version 2.4). All MS/MS samples were analyzed using Mascot (Matrix Science, London, UK; version 2.3). Mascot was set up to search the UniProt *Bacillus* database assuming the digestion enzyme trypsin. Mascot was searched with a fragment ion mass tolerance of 0.020 Da and a parent ion tolerance of 10.0 PPM. MMTS of cysteine and iTRAQ 8-plex of lysine and the n-terminus were specified in Mascot as fixed modifications. Oxidation of methionine and iTRAQ 8-plex of tyrosine were specified in Mascot as a variable modification. The percolator algorithm was used to control protein level false discovery rates (FDR) lower than 1%.

### 4.8. Molecular Docking of FliC

According to the FITC fluorescent label, it can be assumed that rm-FliC acts on the cell membrane and inside of *B. nematocida* B16. Therefore, we preferentially screened the membrane proteins and kinases in iTRAQ and compared them with the whole genome of *B. nematocida* B16 to obtain their amino acid sequences. The three-dimensional structures were retrieved from the Protein Data Bank (PDB) and optimized using Discovery Studio. After removing bound water, hydrogen atoms, and other irrelevant molecules, polar hydrogen atoms were added after processing. The protein-to-protein docking was calculated using ZDOCK 3.0.2 to perform subsequent optimizations on the structure of the first complex in the output prediction complex.

## Figures and Tables

**Figure 1 pathogens-11-00082-f001:**
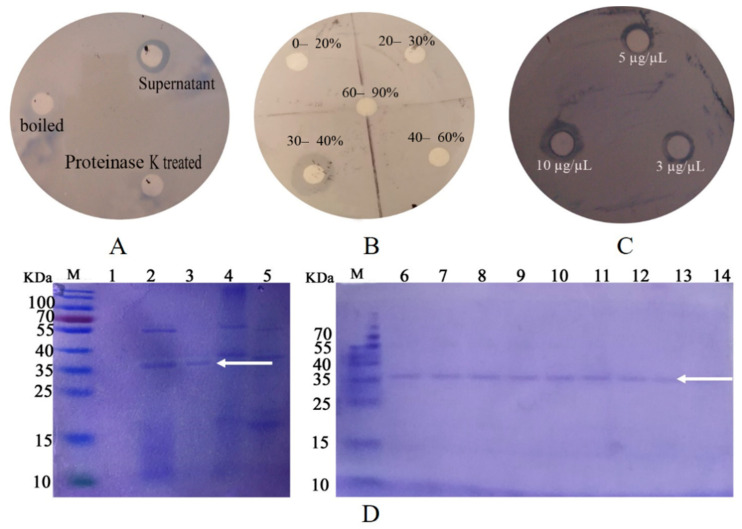
Results of disk diffusion assays and purification process of active protein. (**A**) The effect of supernatants on B16 growth inhibition; (**B**) the effect of different concentration gradient fractions on B16 growth inhibition; (**C**) the effect of different concentration purified protein on B16 growth inhibition; (**D**) SDS-PAGE of the purification process of active protein. Lanes: M, protein marker; 1, fractions of 0–20% ammonium sulfate fractional precipitation; 2, fractions of 20–30% ammonium sulfate fractional precipitation; 3, fractions of 30–40% ammonium sulfate fractional precipitation; 4, fractions of 40–60% ammonium sulfate fractional precipitation; 5. fractions of 60–90% ammonium sulfate fractional precipitation. 6–14: purified protein. The arrow points to the target protein band.

**Figure 2 pathogens-11-00082-f002:**
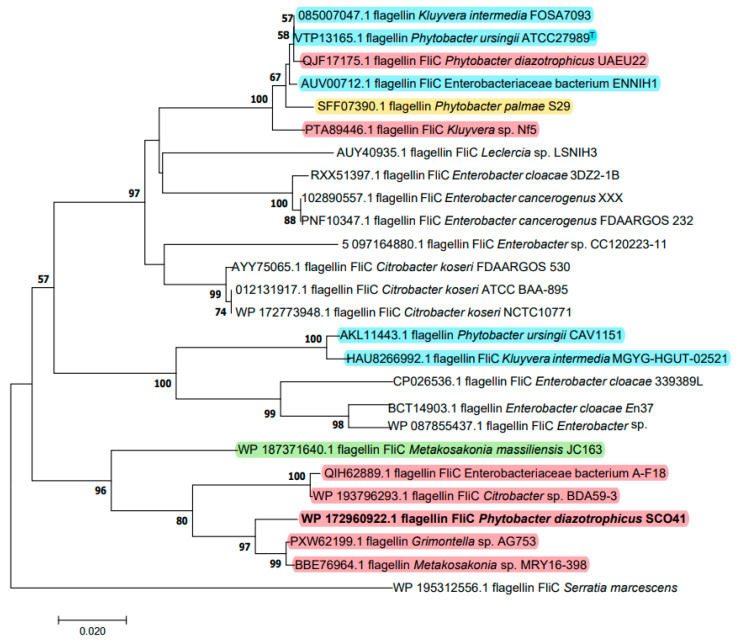
Neighbor-joining unrooted tree of FliC based on amino acid sequences. The optimal tree with the sum of branch length = 0.76278498 is shown. The evolutionary distances were computed using the Poisson correction method and are in the units of the number of amino acid substitutions per site (bar, 2% amino acid substitutions). Bootstrap values of above 50% are given at nodes based on 1000 replications. Sequences belonging to *Phytobacter* species are highlighted in different colors: red, *P. diazotrophicus*; blue, *P. ursingii*; yellow, *P. palmae*; green, *P. massiliensis*. Sequence names were retrieved from Genbank and do not necessarily correspond with the species identification according to genome data.

**Figure 3 pathogens-11-00082-f003:**
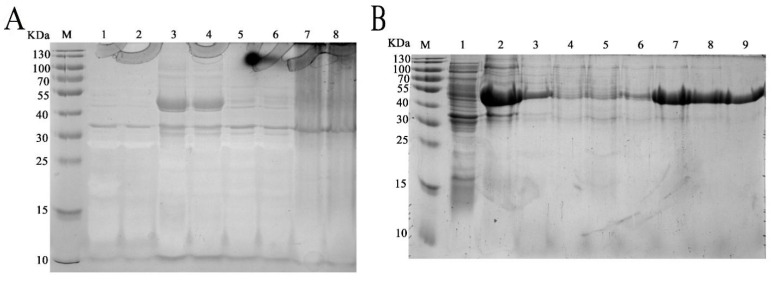
SDS-PAGE (12%) demonstrated the expression of the *p1* in *E. coli* transformants. (**A**) Optimization of induction conditions for heterologous expression: Lanes: M, protein marker; 1, PET32a-BL21; 2, P1-PET32a-BL21without induction; 3, culture precipitate of the transformant P1-PET32a-BL21 with 0.6 mM IPTG at 25 °C for 12 h; 4, the culture supernatant of the transformant of samples in line 3; 5, culture precipitate of the transformant P1-PET32a-BL21 with 0.6 mM IPTG at 30 °C for 12 h; 6, the culture supernatant of the transformant of samples in line 5; 7, culture precipitate of the transformant P1-PET32a-BL21 with 1.0 mM IPTG at 30 °C for 12 h; 8, the culture supernatant of the transformant of samples in line 7. (**B**) Purification process of re-P1: Lanes: M, protein marker; 1, protein fractions before Ni-NTA affinity chromatography; 2, Effluent fractions; 3, fractions of flow through solution in affinity chromatography; lines 4–9: fractions of elution peak with imidazole concentration of 40, 60, 100, 200, 300 and 500 mM in sequence.

**Figure 4 pathogens-11-00082-f004:**
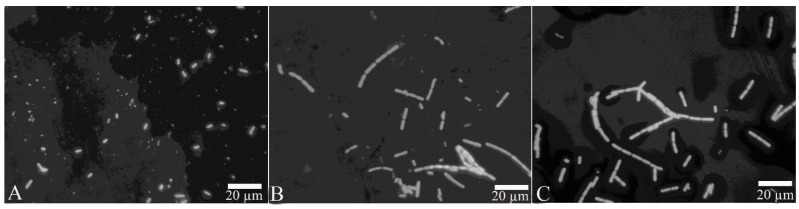
Observation of cell morphology change after treatment with the protein FliC and rm-FliC: (**A**) normal *B. nematocida* B16 cell morphology treated with PBS; (**B**) cell morphology of *B. nematocida* B16 treated with purified FliC; (**C**) cell morphology of *B. nematocida* B16 treated with purified rm-FliC.

**Figure 5 pathogens-11-00082-f005:**
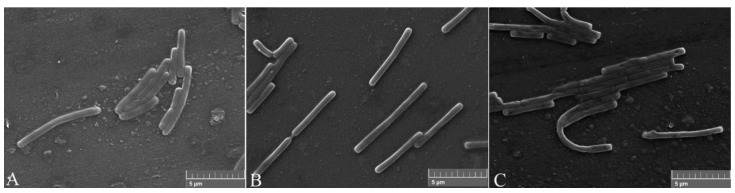
Scanning electronic microscopy (SEM) observation of cell morphology change after treated with the protein FliC and rm-FliC: (**A**) normal *B. nematocida* B16 cell morphology treated with PBS; (**B**) cell morphology of *B. nematocida* B16 treated with purified FliC; (**C**) cell morphology of *B. nematocida* B16 treated with purified rm-FliC.

**Figure 6 pathogens-11-00082-f006:**
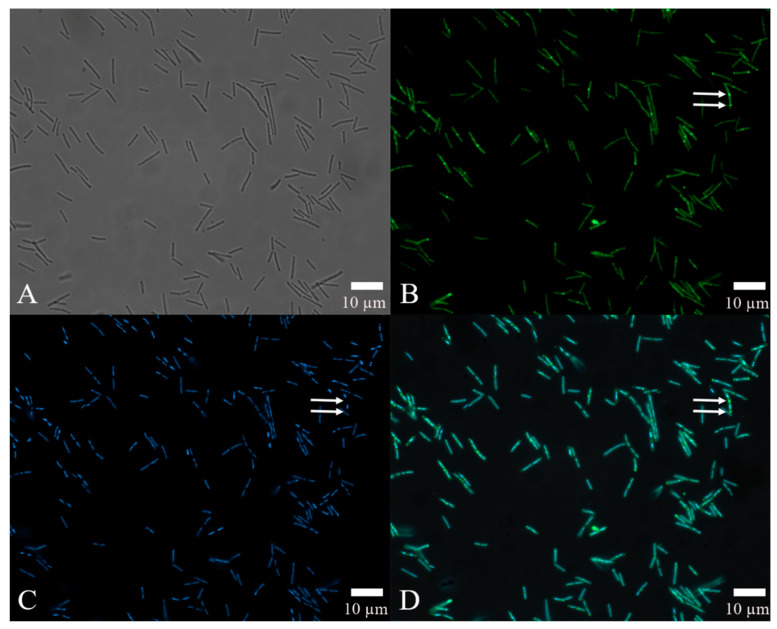
Localization of the protein rm-FliC in B16 cells: (**A**) cell morphology of bacteria under visible light; (**B**) observation on the localization of FITC-FliC in *B. nematocida* B16 at 492 nm; (**C**) observation of the localization results of DAPI staining in *B. nematocida* B16 at 358 nm; (**D**) an overlay image treated with the double-localization of (**B**,**C**). Notes: The arrows in B, C and D refer to the strong fluorescence intensity at the cell division.

**Figure 7 pathogens-11-00082-f007:**
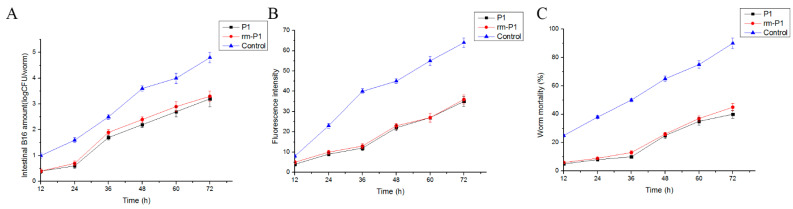
Results of “colonization-resistance” activity assays of purified proteins. (**A**) Quantities of *B. nematocida* B16 bacteria in intestines of the worms after contact with proteins FliC and rm-FliC. (**B**) Fluorescence intensities of the strain B16 in intestines of the worms after contact with proteins FliC and rm-FliC. (**C**) Killing of the worms after contact with proteins FliC and rm-FliC.

**Figure 8 pathogens-11-00082-f008:**
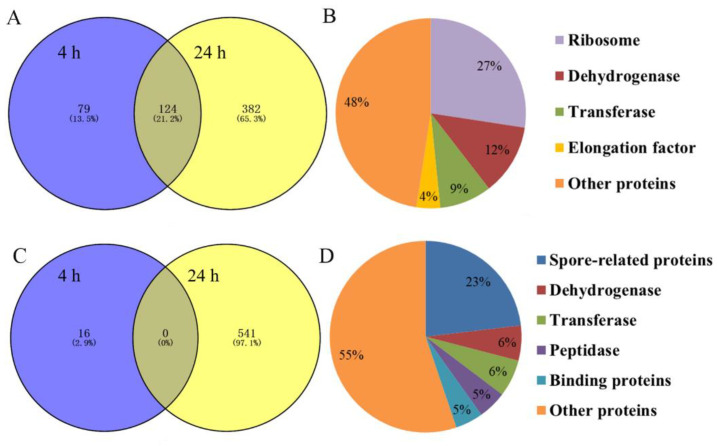
(**A**) Venn diagrams showing the distribution of up-regulated proteins in *B. nematocida* B16 fraction altered by rm-FliC treatment; (**B**) results of up-regulated protein classification and proportion in *B. nematocida* B16 after 24 h of rm-FliC treatment; (**C**) Venn diagrams showing the distribution of down-regulated proteins in *B. nematocida* B16 fraction altered by rm-FliC treatment; (**D**) results of down-regulated protein classification and proportion in *B. nematocida* B16 after 24 h of rm-FliC treatment.

**Figure 9 pathogens-11-00082-f009:**
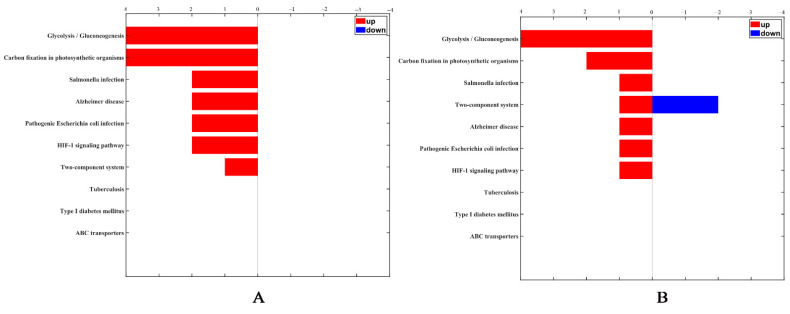
KEGG categorization of differentially expressed proteins in the *B. nematocida* B16 treated with FliC for 4 h (**A**) and 24 h (**B**).

**Figure 10 pathogens-11-00082-f010:**
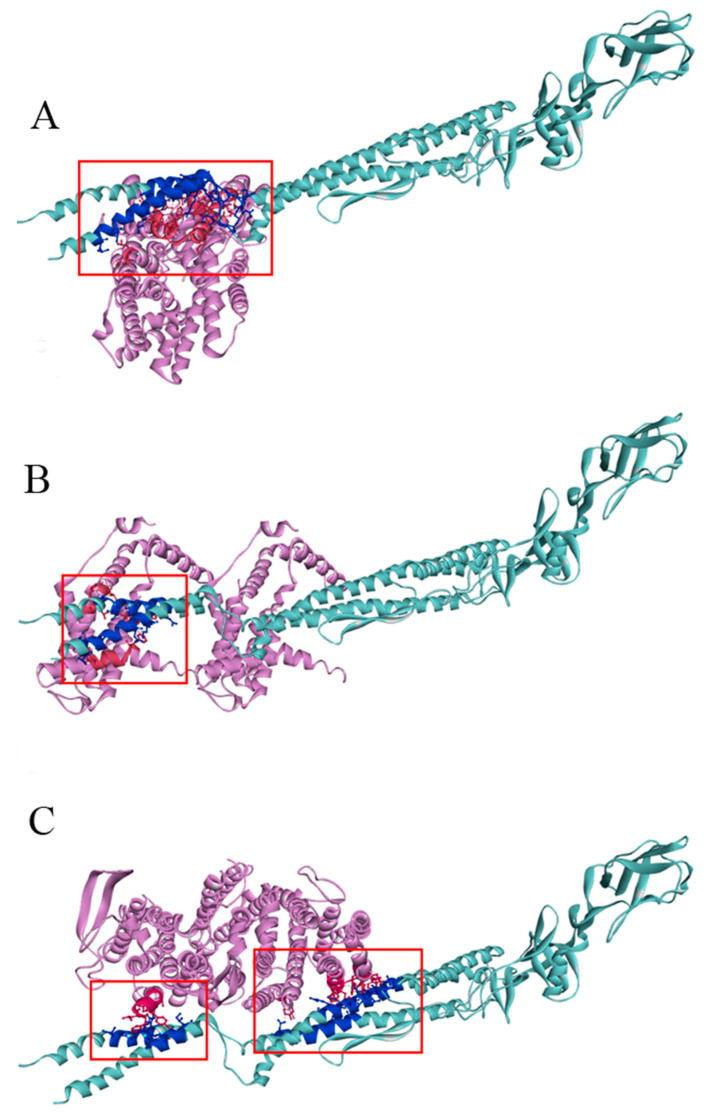
Complex model of three potential receptors of serine/threonine protein kinase (**A**), membrane protein insertase YidC (**B**) and micro-aerobic cytochrome bd terminal oxidase CydB docking with FliC (**C**). The binding sites are shown with ball and stick. The interacting amino acid residues at the binding interface of the complex model are marked in different hues of the corresponding color. Light blue: FliC; dark blue: amino acid residues at the FliC binding interface; light pink: the potential receptor proteins; dark pink: amino acid residues at the receptors’ binding interface.

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
