# Peer review of "Phytobacter diazotrophicus* from Intestine of *Caenorhabditis elegans* Confers Colonization-Resistance against *Bacillus nematocida* Using Flagellin (FliC) as an Inhibition Factor"

_pathogens, 2022, doi:10.3390/pathogens11010082_

Round 1

Reviewer 1 Report

This is a very well written manuscript which brings quite interesting results. 

Very minor corrections are needed:

1- In figure 5, the caption mention twice the Letter C, instead of B and C. Please correct.

2- About the misnaming of Phytobacter on lines 292 through 305, I sugestão to include one extra phrase, considering the article by Ma te al. 2020 - https://doi.org/10.1007/s00284-020-02004-4, that clarifies many of the misleading names that appears in Figure 2 and also suggests that Metakosakonia should be substitutes by Phytobacter.  

3- The bacterial taxonomy names at the reference section all have the genus name in lowercase, instead of uppercase. Please review.

Author Response

1- In figure 5, the caption mention twice the Letter C, instead of B and C. Please correct.

Answer: We corrected the caption in the figure. Thank you so much for your comments!

2- About the misnaming of Phytobacter on lines 292 through 305, I sugestão to include one extra phrase, considering the article by Ma te al. 2020 - https://doi.org/10.1007/s00284-020-02004-4, that clarifies many of the misleading names that appears in Figure 2 and also suggests that Metakosakonia should be substitutes by Phytobacter.  

Answer: We gave an explanation to the misnaming of Metakosakonia in the revision. Thank you so much for your comments!

  • The bacterial taxonomy names at the reference section all have the genus name in lowercase, instead of uppercase. Please review.

Answer: We corrected the names in the all references in the revision. Thank you so much for your comments!

Reviewer 2 Report

The paper by Niu et al. describes the characterization of flagellin (FliC) as a bacterial factor that potentially affects colonization of the C. elegans intestine by Bacillus nematocida. While the results suggest an exciting mechanism by which bacteria could inhibit the growth or colonization of other microbes, the conclusions are not supported by the results. In the current state, the manuscript is not suitable for publication. Some of the major concerns include the following:  

-The authors demonstrated using a disk diffusion assay that spent supernatant from P. diazotrophicus SCO41 inhibits the growth of B. nematocidal B16 and that FliC is the active component. However, purified FliC did not exhibit any inhibitory effect on B16. These results indicate that it is not FliC that is the active inhibitory component.

-Growth inhibition by SCO41 and its supernatant should also be tested in axenic culture

-Did the authors consider that antimicrobial peptides or peptide antibiotics could play a role in the observed inhibition (disk diffusion)?

-What is the evidence that FliC is secreted?

-The paper would benefit from having additional experiments that confirm the property of the active inhibitory fraction. For example, a proteinase k digestion of the spent supernatants.

-The authors describe the isolation of 52 isolates, but no screening results are presented

-The conclusions that FliC is localized in the cytoplasm are not evident from the provided data. It is also not clear what antibody was used to detect FliC

-The results stating that DAPI helped confirm that the protein FliC does act on bacterial cells are not clear. It is also not clear that the protein localizes at the cell division site

-It is not clear what antibody was used and whether the antibody used is actually specific to FliC. The authors should use a his-tag antibody as well

-I don’t understand why the cells were treated with FliC for up to 24h. What is the doubling time? Are these cells in the stationary phase? I have a feeling that the proteomic analysis is comparing

-The authors conclude that FliC may inhibit stress responses in B16. To conclude that, stress tolerance assays should be performed (i.e., Heat, H2O2, etc.).

Minor comments

-The authors state that a number of gut bacteria were screened for the inhibitory property. What bacteria? The screen is not described in the results section

-use arrows in Fig 1

-Is the spent supernatant also inhibitory to other gram-positive bacteria (ex. B. subtilis)? Or gram-negative E. coli?

-Avoid referencing any unpublished data

-The authors use a recombinant (rm-FliC) and non-recombinant(?) FliC. It is not clear where did the untagged FliC come from.

-The study would benefit from co-colonization experiments (B16 and SCO41) to determine whether SCO41 inhibits colonization of B16.  

-The rationale and results of the molecular docking analysis are not clear

-The methods section is not written with sufficient details to replicate the study

-Please proofread the paper for grammatical errors.

-how were the worms ground? (line 95)

-line 18: write out SEM

-line 34: consists of

-line 44: [8] not italic

-how were bacteria identified (line 96)?

-what is E. coli 109g?

-the authors state the worms were cultured on food for 24h until L4. C. elegans do not reach L4 after 24h. (line 114)

-why was water and not M9 used to wash worms? Line 118

-Write 50 numerical (line 120)

-line 307: double space

-Figure 3 is not labeled (A and B)

-Figure 4 needs scale bars

-Figure 5 labeling A-C needs to be consistent between figures. Remove the bottom part of the panels, add scale bars

-Figure 6 needs scale bars and labels A-D need to be consistent with other figures

-Figure 7 legend: Fluorescent intensity of what? …after contact with proteins – what proteins?

-lines 545-546 italic names

-

Author Response

-The authors demonstrated using a disk diffusion assay that spent supernatant from P. diazotrophicus SCO41 inhibits the growth of B. nematocidal B16 and that FliC is the active component. However, purified FliC did not exhibit any inhibitory effect on B16. These results indicate that it is not FliC that is the active inhibitory component.

Answer: Thank you for your thoughtful comments. Actually, we did test the inhibitory effect on B16 of the purified FliC both in vitro and in vivo. The results showed that the purified FliC had a significant inhibition on the growth of B16 in vitro and colonization resistance activity against B16 in vivo. We have complemented the experimental results in the revised article.

-Growth inhibition by SCO41 and its supernatant should also be tested in axenic culture

Answer: Thank you for your thoughtful comments. The test results on the growth inhibition inactivity by SCO41 and its supernatant have been compiled and published in our previous  article (Wang et al., Whole-genome analysis of the colonization-resistant bacterium Phytobacter sp. SCO41T isolated from Bacillus nematocida B16-fed adult Caenorhabditis elegans. Molecular Biology Reports. 2019. 46: 1563-1575). Thanks for your suggestions again!

-Did the authors consider that antimicrobial peptides or peptide antibiotics could play a role in the observed inhibition (disk diffusion)?

Answer: Yes, we did. Antimicrobial peptides or peptide antibiotics are also reported as common antimicrobial factors. We found the flagellin via tracking the inhibition activities of the fermentation supernatant. No antimicrobial peptides were found during the preliminary analysis of the bacterial genome and the activity tracking of the fermentation broth. Whether there are antimicrobial peptides in the bacteria SCO41 is still under investigation. Thanks a lot for your kind reminding.

-What is the evidence that FliC is secreted?

Answer: The protein FliC was traced and obtained in the supernatant after fermentation, and the bacteria was not broken, so it was secreted. Furthermore, there are some literatures reported FliC being secreted in the extracellular medium such as Sanna Nyström et al., Vaccinces, 2013, 1, 415-443 (doi:10.3390/vaccines1040415) and Chandrabali Ghose et al., 2016, 5, e8 (doi:10.1038/emi.2016.8). Thank you for your question.

-The paper would benefit from having additional experiments that confirm the property of the active inhibitory fraction. For example, a proteinase k digestion of the spent supernatants.

Answer: Yes, 0.1 M protease K added to the spent supernatants were used as controls. Thank you for your comments very much!

-The authors describe the isolation of 52 isolates, but no screening results are presented

Answer: Yes, the screening results were presented in our published article (Wang et al., Whole-genome analysis of the colonization-resistant bacterium Phytobacter sp. SCO41T isolated from Bacillus nematocida B16-fed adult Caenorhabditis elegans. Molecular Biology Reports. 2019. 46: 1563-1575). Thanks for your suggestions again!

-The conclusions that FliC is localized in the cytoplasm are not evident from the provided data. It is also not clear what antibody was used to detect FliC

Answer: The purified FliC is a recombinant protein heterologously expressed using pET-32a in E. coli and it is labeled with His-tag. The his-tag antibody was used to specific to FliC.

The localization experiments demonstrated that the protein FliC is localized on the whole cell. The results of automated fluorescence quantitative analysis using Image J showed that the fluorescence intensity was uniformly distributed throughout the cell. Some cells were found to show significantly enhanced fluorescence intensity at the division point ends, as the arrows indicated in Fig. 6B. DAPI helped to confirm that the protein FliC does act on the chromosome of the bacterial cells (Fig. 6C). After the two images overlayed, it was found that the protein FliC showed a very strong fluorescence intensity at the cell division in the B16 cells (Fig. 6D). According to the results above, it can be inferred that the protein FliC may act both the cell membrane and cytoplasm of B16 cells.

Thanks a lot for the reviewer’s comments!

-The results stating that DAPI helped confirm that the protein FliC does act on bacterial cells are not clear. It is also not clear that the protein localizes at the cell division site

Answer: Thank you for the reviewer’s comments! We have corrected the description in the revised manuscript. 

DAPI helped confirm that the protein FliC does act on the chromosome of the bacterial cells. Quantitative analysis based on the fluorescence intensity showed that strong fluorescence values were displayed at the cell division site, and it was inferred that the protein FliC may act on the cell division site. It was indicated by an arrow in the figure 6 in the revised manuscript.

-It is not clear what antibody was used and whether the antibody used is actually specific to FliC. The authors should use a his-tag antibody as well

Answer: Thank you for the reviewer’s comments! Actually, we indeed used the his-tag antibody.

-I don’t understand why the cells were treated with FliC for up to 24h. What is the doubling time? Are these cells in the stationary phase? I have a feeling that the proteomic analysis is comparing

Answer: Thank you for the reviewer’s comments! By measuring the OD value of the growth of the bacteria B16 after the protein FliC treatment, it was found that FliC displayed the most significant inhibitory effect against B16 growth at 24 h. At 48 h, the differences of B16 growth between FliC-treated strains and the controls showed decreased. The cells are already in the stationary phase at this time. Therefore, the samples treated for 24 h were selected for proteomics analysis.

-The authors conclude that FliC may inhibit stress responses in B16. To conclude that, stress tolerance assays should be performed (i.e., Heat, H2O2, etc.).

Answer: As the reviewer suggested, we performed the stress tolerance assays experiments. The results demonstrated that FliC-treated strains exhibited lower tol-erance of heat, dry, H2O2 than the control B16 strains exhibited. Thanks a lot for the comments!

Minor comments

-The authors state that a number of gut bacteria were screened for the inhibitory property. What bacteria? The screen is not described in the results section

Answer: We complemented the screen results in the revised manuscript.

-use arrows in Fig 1

Answer: We added arrows in the figure, thank you very much for your comments!

-Is the spent supernatant also inhibitory to other gram-positive bacteria (ex. B. subtilis)? Or gram-negative E. coli?

Answer: Yes, the spent supernatant of the strain SCO41 showed inhibitory to B. subtilis but not to E. coli.

-Avoid referencing any unpublished data

Answer: OK. We deleted the reference unpublished data. Thanks for reminding.

-The authors use a recombinant (rm-FliC) and non-recombinant(?) FliC. It is not clear where did the untagged FliC come from.

Answer: The untagged FliC comes from the spent supernatant after SCO41 fermentation. We provided the explanation in the revised manuscript. Thank you for your thoughtful comments!

-The study would benefit from co-colonization experiments (B16 and SCO41) to determine whether SCO41 inhibits colonization of B16.  

Answer: Thank you for your great suggestions! You provided us with another model to test colonization resistance. In the next step, we will label the tested strain with red fluorescence protein (rfp) or yellow fluorescence protein (yfp), plus with B16g which is labeled with green fluorescent protein (gfp) to test the colocalization experiment.

-The rationale and results of the molecular docking analysis are not clear

Answer: Thank you for your thoughtful comments!

We have corrected the description and the figure in the revised manuscript. Also, we put the detail amino acid residue sequences of the interactions between FliC and the three receptors in Supplemental tables 1-4. Thanks again for your comments!

-The methods section is not written with sufficient details to replicate the study

Answer: Thank you for your thoughtful comments! We improved the description of the method section in the revised manuscript.

-Please proofread the paper for grammatical errors.

Answer: We corrected the grammatical errors in the revised manuscript.

-how were the worms ground? (line 95)

Answer: We described the method in detail in the revised manuscript. Thanks a lot!

-line 18: write out SEM

Answer: We wrote out SEM in the revision. Thanks a lot!

-line 34: consists of

Answer: We corrected it. Thank you for your careful comments!

-line 44: [8] not italic

Answer: We corrected it. Thank you for your careful comments!

-how were bacteria identified (line 96)?

Answer: We complemented it. Thank you for your careful comments!

-what is E. coli 109g?

Answer: E. coli 109g is the E. coli strain labeled with green fluorescent protein as described in the literature (Niu et al., 2010, supplements). Thank you for your question!

-the authors state the worms were cultured on food for 24h until L4. C. elegans do not reach L4 after 24h. (line 114)

Answer: Sorry for the unclear and inaccurate description. In fact, we selected the nematodes that grow to the L4 stage for tests. We changed the inappropriate description in the revision. Thanks a lot for your reminding!

-why was water and not M9 used to wash worms? Line 118

Answer: Yes, M9 buffer is usually used to wash worms. Here we used sterile water based on the literature. Thanks a lot for your kind comments!

-Write 50 numerical (line 120)

Answer: We wrote 50 numerical as the reviewer suggested. Thank you for your carefulness!

-line 307: double space

Answer: We adjusted the line space. Thank you so much for your carefulness!

-Figure 3 is not labeled (A and B)

Answer: We labeled (A and B) in the figure. Thank you so much for your carefulness!

-Figure 4 needs scale bars

Answer: We added the scale bars in the figure. Thank you so much for your carefulness!

-Figure 5 labeling A-C needs to be consistent between figures. Remove the bottom part of the panels, add scale bars

Answer: We improved the figure as the reviewer suggested. Thank you so much for your carefulness!

-Figure 6 needs scale bars and labels A-D need to be consistent with other figures

Answer: We improved the figure as the reviewer suggested. Thank you so much for your carefulness!

-Figure 7 legend: Fluorescent intensity of what? …after contact with proteins – what proteins?

Answer: We improved the figure legend in the revision. Thank you so much for your carefulness!

-lines 545-546 italic names

Answer: We corrected the names in the all references in the revision. Thank you so much for your comments!

Round 2

Reviewer 2 Report

Thank you for addressing most of my comments. I do have additional concerns and suggestions:

-Please change protease k to proteinase k throughout the manuscript, including Fig 1.

-Line 286: remove “strong”. It does not look like the inhibition is strong

-Fig 1: please move labels (A-D) to the top-left side of each panel. This actually applies to all figures.

-Line 287: remove “almost” Fig 1A shows no zone of inhibition for PK-treated and boiled samples

-Lines 292-293: FliC is described as having inhibitory activity before mass spec results are revealed which identify FliC. This has to be revised so there is a logical flow.

-Fig 1 legend needs to be revised: the authors describe the inhibitory activity of supernatants and purified FliC in terms of resistance. For example, instead of writing “Resistance of supernatants…” it should state the effect of supernatants on B16 growth inhibition or something similar. The same applies to Fig1C legend. Additionally, please remove % from the first number by changing the fractions to 30-40%, etc…

From review no. 1:

-Growth inhibition by SCO41 and its supernatant should also be tested in axenic culture

“Answer: Thank you for your thoughtful comments. The test results on the growth inhibition inactivity by SCO41 and its supernatant have been compiled and published in our previous article (Wang et al., Whole-genome analysis of the colonization-resistant bacterium Phytobacter sp. SCO41T isolated from Bacillus nematocida B16-fed adult Caenorhabditis elegans. Molecular Biology Reports. 2019. 46: 1563-1575). Thanks for your suggestions again!”

-The above reference does not describe SCO41-mediated growth inhibition of B16 in vitro

-line 350: add space between 500 and mM

-Explain what the arrows are in the Fig 6 legend.

-line 373: what do the authors mean by writing that DAPI staining confirmed the position of the cells?

-line 376: these data suggest

From review no. 1:

-The conclusions that FliC is localized in the cytoplasm are not evident from the provided data. It is also not clear what antibody was used to detect FliC

Answer: The purified FliC is a recombinant protein heterologously expressed using pET-32a in E. coli and it is labeled with His-tag. The his-tag antibody was used to specific to FliC.

The localization experiments demonstrated that the protein FliC is localized on the whole cell. The results of automated fluorescence quantitative analysis using Image J showed that the fluorescence intensity was uniformly distributed throughout the cell. Some cells were found to show significantly enhanced fluorescence intensity at the division point ends, as the arrows indicated in Fig. 6B. DAPI helped to confirm that the protein FliC does act on the chromosome of the bacterial cells (Fig. 6C). After the two images overlayed, it was found that the protein FliC showed a very strong fluorescence intensity at the cell division in the B16 cells (Fig. 6D). According to the results above, it can be inferred that the protein FliC may act both the cell membrane and cytoplasm of B16 cells.

-Colocalization does not confirm physical interaction. It’s only a prediction.

From Review No. 1:

-The authors conclude that FliC may inhibit stress responses in B16. To conclude that, stress tolerance assays should be performed (i.e., Heat, H2O2, etc.).

Answer: As the reviewer suggested, we performed the stress tolerance assays experiments. The results demonstrated that FliC-treated strains exhibited lower tol-erance of heat, dry, H2O2 than the control B16 strains exhibited. Thanks a lot for the comments!

-the results are missing from the manuscript

-the authors are strongly advised to carefully proofread the manuscript and correct any additional grammar errors.

Author Response

To the reviewer (Round 2):

-Please change protease k to proteinase k throughout the manuscript, including Fig 1.

Answer: we changed protease k to proteinase k as the reviewer suggested. Thank you for your comment.

-Line 286: remove “strong”. It does not look like the inhibition is strong

Answer: we removed “strong”as the reviewer suggested. Thank you for your comment.

-Fig 1: please move labels (A-D) to the top-left side of each panel. This actually applies to all figures.

Answer: we moved labels (A-D) to the top-left side of each panel in all figures except figures 4, 5, 6. We think the labels (A-D) would be better to be consistent with the scale bar, thanks a lot for your kind reminding.

-Line 287: remove “almost” Fig 1A shows no zone of inhibition for PK-treated and boiled samples

Answer: we removed “almost” as the reviewer suggested. Thank you for your comment.

-Lines 292-293: FliC is described as having inhibitory activity before mass spec results are revealed which identify FliC. This has to be revised so there is a logical flow.

Answer: Sorry for our neglect. We changed FliC as protein. Thank you for your carefulness.

-Fig 1 legend needs to be revised: the authors describe the inhibitory activity of supernatants and purified FliC in terms of resistance. For example, instead of writing “Resistance of supernatants…” it should state the effect of supernatants on B16 growth inhibition or something similar. The same applies to Fig1C legend. Additionally, please remove % from the first number by changing the fractions to 30-40%, etc.

Answer: As the reviewer suggested, we revised the Fig 1 legend. Thank you so much for your comment!

From Review No. 1:

-Growth inhibition by SCO41 and its supernatant should also be tested in axenic culture

Answer: Thank you for your thoughtful comments. The test results on the growth inhibition inactivity by SCO41 and its supernatant have been compiled and published in our previous article (Wang et al., Whole-genome analysis of the colonization-resistant bacterium Phytobacter sp. SCO41T isolated from Bacillus nematocida B16-fed adult Caenorhabditis elegans. Molecular Biology Reports. 2019. 46: 1563-1575). Thanks for your suggestions again!”

-The above reference does not describe SCO41-mediated growth inhibition of B16 in vitro

Answer: Thank you for your carefulness! The tests on SCO41-mediated growth inhibition of B16 in vitro was performed during the initial screening process. After confirming that the strains can inhibit B16 colonization in vivo and inhibiting B16 growth in vitro, we selected the  strain SCO41 with the strongest activities as the target material in following research. The result of the SCO41-mediated growth inhibition of B16 in vitro was as the following figure:

-line 350: add space between 500 and mM

Answer: We added the space. Thank you so much for your comment!

-Explain what the arrows are in the Fig 6 legend.

Answer: We complemented the description in the Fig 6 legend. Thanks a lot!

-line 373: what do the authors mean by writing that DAPI staining confirmed the position of the cells?

Answer: DAPI is a fluorescent stain that binds to DNA. The initial purpose of using DAPI was to help confirm the location on the bacterial cell instead of other impurities, and then use FITC to label the protein FliC location. Thank you so much for your comment!

-line 376: these data suggest

Answer: Sorry for the mistake. We have removed “s”. Thanks a lot!

From Review No. 1:

-The conclusions that FliC is localized in the cytoplasm are not evident from the provided data. It is also not clear what antibody was used to detect FliC

Answer: The purified FliC is a recombinant protein heterologously expressed using pET-32a in E. coli and it is labeled with His-tag. The his-tag antibody was used to specific to FliC.

The localization experiments demonstrated that the protein FliC is localized on the whole cell. The results of automated fluorescence quantitative analysis using Image J showed that the fluorescence intensity was uniformly distributed throughout the cell. Some cells were found to show significantly enhanced fluorescence intensity at the division point ends, as the arrows indicated in Fig. 6B. DAPI helped to confirm that the protein FliC does act on the chromosome of the bacterial cells (Fig. 6C). After the two images overlayed, it was found that the protein FliC showed a very strong fluorescence intensity at the cell division in the B16 cells (Fig. 6D). According to the results above, it can be inferred that the protein FliC may act both the cell membrane and cytoplasm of B16 cells.

-Colocalization does not confirm physical interaction. It’s only a prediction.

Answer: Sorry for the comment. We have improved the relevant description. Thanks a lot!

From Review No. 1:

-The authors conclude that FliC may inhibit stress responses in B16. To conclude that, stress tolerance assays should be performed (i.e., Heat, H2O2, etc.).

Answer: As the reviewer suggested, we performed the stress tolerance assays experiments. The results demonstrated that FliC-treated strains exhibited lower tolerance of heat, dry, H2O2 than the control B16 strains exhibited. Thanks a lot for the comments!

-the results are missing from the manuscript

Answer: Thank you for your reminding. We described the results on the stress tolerance assays experiments in the revised manuscript and complemented the data in supplements.

-the authors are strongly advised to carefully proofread the manuscript and correct any additional grammar errors.

Answer: Thank you for your reminding. We have carefully proofread the manuscript and corrected the grammar errors in the whole manuscript.